# The Association between Social Participation and Loneliness of the Chinese Older Adults over Time—The Mediating Effect of Social Support

**DOI:** 10.3390/ijerph19020815

**Published:** 2022-01-12

**Authors:** Lijuan Zhao, Lin Wu

**Affiliations:** School of Sociology, Wuhan University, Wuhan 430072, China; 2020101170007@whu.edu.cn

**Keywords:** loneliness, social participation, social support, Hierarchical Linear Modeling

## Abstract

Based on activity theory, this paper employed data from the 2013, 2015, and 2018 waves of the China Health and Retirement Longitudinal Survey, and adopted Hierarchical Linear Modeling and longitudinal mediation analysis to explore the temporal variation characteristics of loneliness and the influence of social participation on loneliness in Chinese Older Adults, as well as the mechanism of them. The study found that loneliness among older adults overall was at a moderate level from 2013 to 2018 and increased over time, which may be related to decreasing social participation from year to year. Decreased social participation was associated with increased loneliness over time (β = −0.060, *p* < 0.001) and lower social support (β = 0.109, *p* < 0.001), which was associated with more loneliness (β = −0.098, *p* < 0.001). In addition, social support played a significant mediating role in the realization of social participation in alleviating loneliness. Social participation can not only directly reduce loneliness, but also reduce loneliness by increasing social support.

## 1. Introduction

China is currently experiencing a significantly changing population, with the decline in fertility rate and the extension of life expectancy leading to a substantial increase in the older adults population. It is important to note that China is also experiencing a rapidly developing economy and fast urbanization, both of which have driven young adults’ migration from home and distributed the traditional family structure, resulting in a larger number of non-traditional older adults populations, such as “empty-nester” [1,2] and “floating older adults” [3]. All the above social changes contribute to increase the vulnerability of older adults to experiencing loneliness. The China National Committee on Ageing reported that nearly a third of young-old adults felt lonely, its prevalence in old-old individuals was higher than 50% in China in 2018 [4], and the proportion of the empty-nest group who felt lonely was even 78.1% [1]. Loneliness not only threatens the physical and mental health of older adults, but also undermines social harmony, which should be taken seriously.

Loneliness is a psychological experience that changes constantly over time [5,6,7,8]. Two cross-temporal meta-analysis studies in the literature investigated changes in Chinese older adults’ loneliness through correlating loneliness scores with several social indicators, including urbanization level, divorce rate, and unemployment rate, and revealed a birth cohort increase in loneliness levels in Chinese older adults [6,9]; that is to say, the level of loneliness among later-birth cohorts of Chinese older adults was higher than those earlier-birth cohorts. In parallel, evidence from systematic reviews and empirical research indicates that loneliness can be prevented or relieved by interventions such as increased social engagement and social contacts [10,11,12]. Taken together with the serious situation that both the prevalence of loneliness among older Chinese now and the proportion of empty-nesters and floating older adults will increase continuously in the future [13], it is vital and valuable to have a comprehensive understanding of the temporal nature and mechanism of loneliness among older adults in China, a culture that emphasizes older people first and filial piety. There exists abundant theoretical and empirical studies focusing on the factors related to and consequences of loneliness in terms of cross-sectional perspectives [6,8,9], but few research studies based on longitudinal and large-scale data have considered the time-varying characteristics of older adults’ loneliness and predictors or determining factors, especially for Chinese older adults in the last 10 years, a period where dramatic social changes are taking place [13]. Therefore, it is vital to understand the temporal trend of loneliness among Chinese older adults in this last decade and to distinguish cross-sectional loneliness (a survey point) and longitudinal loneliness (the temporal property of loneliness). Comprehensive knowledge of cohort differences and age differences in loneliness is fundamental to make targeted strategies to ameliorate their symptoms, such as encouraging social participation.

### 1.1. Social Participation and Loneliness

Activity theory proposes that successful aging requires the interests and activities engaged in in mid-adulthood to be continued and the prevention of decreases in the number and type of social interactions [14], emphasizing the importance of social engagement. According to the social integration perspective, people change their roles in different periods over their lifetime to maintain social integration, obtain social capital and increase their quality of life [15]. With aging and retirement, the social value of older adults seems to decrease and their social network shrinks; therefore, participating in social activities may be more necessary than ever. Social participation can not only provide social relationships and resources to make up for their loss [16], but also build new social networks to expand their existing ones, which are a prerequisite for lower loneliness levels [17,18,19].

The existing research studies have explored the influence of social participation on loneliness from both theoretical and empirical angles [17,18,20]. Niedzwiedz et al. (2016) analyzed the relationship between social participation and loneliness with a sample of 29,795 older people, and found that those older adults who often took part in social activities were less likely to be lonely [17]. Similarly, intervention studies have shown that interventions that increase social participation can effectively increase the social support that older adults receive, thus significantly reducing their loneliness [18,19,21]. Some longitudinal research studies show that the persistence or intensification of loneliness among older adults is largely related to the decrease in social participation due to aging [22]. Despite the abundant academic research studies on the association between social participation and loneliness among Western older adults, relevant studies based on Chinese older adults are relatively scarce with a few studies confirming the positive effects of social participation from the perspective of mental health-related fields [23,24].

In addition, continuity theory proposes that people only need to maintain their required frequency of social participation to obtain an optimal effect on their physical and mental health, which indicates that the influence of social participation should be viewed dialectically [14] and highlights individuals’ autonomy and moderate frequency rather than the more the better [25,26]. A study examined the cross-sectional associations of the type, frequency and autonomy of social participation with physical and mental health, and the results showed that only autonomous and appropriately frequent (i.e., weekly, monthly rather than yearly) social participation had a positive impact on mental health [25]. The best effect of participation type and frequency on loneliness may be related to individual characteristics such as age, socio-economic status, and personality traits. Previous studies relevant to personality traits in older adults reported that neuroticism and extraversion were the most influential personality factors [27]. Specifically, for those who are active and extraverted, a high frequency of social participation is conducive to lower loneliness, while for those who suffer from social disorders (i.e., anxiety about talking to others in public) or enjoy being alone, infrequent participation may be better. In sum, it is not difficult to infer that the influence of social participation on loneliness also has differences in participation intention and frequency.

### 1.2. The Mediator: Social Support

The Convoy Model of Social Support explores the effect of social support on older adults’ physical and mental well-being from a lifetime developmental perspective [28], and emphasizes the importance of persons’ social network surrounding them, with more variation in the types of networks indicating more social support that they could receive and a much healthier physical and mental status. In contrast, socioemotional selectivity theory points out that unlike younger people who expand their social network through actively engaging in various social participation events, older adults tend to intentionally shrink their network and select their most valuable and intimate relationships to invest in [29], which may not be appliable to Chinese older adults in current society. Within the context of traditional family structure and collective culture in China, older adults tend to put the family member first, especially children, then friends and peers [23]. However, the last 30 years have seen drastic declines in fertility, diluted filial piety, and uneven rates of economic mobility, all of which have contributed to rapid increases in empty-nest households and in the proportions of left-behind older adults whose adult children leave home for employment. More family fragmentation and smaller family size prevent older adults from receiving family support; thus, they fail to nurture their desired family relationship. In order to compensate for the loss of close family attachment and to alleviate potential loneliness, they attempt to seek alternative sources of support. Hence, social participation becomes one of the most important channels for them to substitute their family network with, a convoy of late-life social new networks [30].

Social support as a protective factor of loneliness has also been widely analyzed and verified [31,32,33,34,35,36]. Meanwhile, some studies suggest that social support is a mediating factor for social participation to improve the mental health and happiness of older adults [37,38,39]. Intervention studies have showed that by encouraging older adults to take an active part in social activities, they gain more peer support and social support; thus, we suggest that social support may be a potential mediator of the association between social participation and loneliness. In addition to social participation patterns (i.e., frequency [25]) and social support sources (i.e., children vs. social network members [23,30]), the association among social participation, social support and loneliness may vary with individual differences, including age, gender, marital status, living arrangement, health status, and economic conditions [1,3,8,9]. For example, some previous studies showed that compared with older adults who live with their children, empty-nest older adults were more likely to benefit from various social activities [1]. Given the relatively higher and continuously increasing prevalence of Chinese older adults’ loneliness and the lack of studies on the mediating role of social support, the current study aims to fill this gap by examining how social participation in the Chinese social context may influence the loneliness trajectories of older people over time.

In a nutshell, although the direct effects of social participation or social support on loneliness among older adults have been widely explored at home and abroad, very few studies have tested these associations using longitudinal and large-scale nationwide samples, which is necessary to draw conclusions on the causal relationship between variables and is also called for in some previous studies [40,41]. Although systematic reviews of interventional studies show that training on both social participation and social support had beneficial effects on loneliness among older adults [10,11,12], which addresses the causal relationships between variables, to our knowledge, there is no relevant research regarding social participation as an interventional strategy to alleviate loneliness among older Chinese adults. Furthermore, interventional studies have shown several limitations, such as small and convenience samples, and a lack of consideration about the dynamic traits of loneliness according to life circumstances and aging. Hence, this present study aims to examine the association between social participation and loneliness among older adults through adopting three waves of data from the CHARLS (2013, 2015 and 2018) and longitudinal analysis models. We also investigated whether social support helped explain the relationship between social participation and loneliness, while controlling for key covariates such as age, gender, marital status, health status and living arrangements.

## 2. Research Questions and Hypothesis

In order to overcome the above shortcomings of previous empirical studies, this study uses a statistical approach that enables the pattern of psychological variables and behavior to be tracked over time to investigate the temporal trajectory of loneliness, the influence of social participation and its mechanism through a longitudinal study design. This paper aims to propose answers to the following three issues:

RQ 1: Does the level of loneliness among Chinese older adults increase over time?

H-1.1: The level of loneliness among the older adults population in the later-born cohort is higher than that of the earlier-born.

RQ 2: Does social participation have a longitudinal effect on loneliness?

H-2.1: From the perspective of longitudinal relationship, social participation has a certain promoting effect on reducing loneliness;

H-2.2: The higher the frequency of social participation, the lower the loneliness level of the older adults.

RQ 3: If social participation does affect loneliness over time, is the relationship mediated by social support?

H-3.1: Social support plays a mediating role in the longitudinal correlation between social participation and loneliness.

## 3. Data and Methods

### 3.1. Data Source

The data for this study come from three waves of the Chinese Health and Retirement Longitudinal Studies (CHALRS) conducted by the Institute of Social Science Survey (ISSS) of Peking University. The CHARLS adopted a multistage stratified probability-proportionate-to-size design to collect a nationally representative sample of Chinese residents aged 45 years and older. We obtained the data from the official website http://charls.pku.edu.cn (accessed on 1 October 2021), which is available to users worldwide.

For the current analysis, the main respondents in 2013 aged 60 years or older were first selected (N1 = 8934). The standard excluded from final analysis of the data was data missing on the key variables (i.e., the loneliness measure). First, if a respondent has a missing item on the loneliness measure, then she/he is excluded from the analysis. Approximately 13% of those excluded from the analysis were missing on the loneliness measure. According to this standard, 1158, 963, and 1426 respondents in 2013, 2015 and 2018, respectively, were excluded from the analysis for this reason, and they are not significantly different from those kept in the analysis in terms of their key demographic information, including gender, age, income, marriage status and education. Then, after further deleting some missing values of key independent variables and covariates, our final sample size was 25,192, which included 7208, 8381, and 9063 respondents in 2013, 2015, and 2018, respectively. There were 12,232 independent respondents, and the proportions of respondents who participated in one wave, two waves, and three waves were 30%, 33%, and 37%, respectively.

Ethical approval for collecting data on human subjects was received from Peking University by their institutional review board. Since we used secondary data with no identifiable information, no formal approval from an institutional review board was required for this study.

### 3.2. Measurements

#### 3.2.1. Loneliness

Loneliness was assessed using a single item from the Centre for Epidemiological Studies scale (CES-D), which assesses the frequency of feeling lonely in the previous week. Respondents were asked to rate the item on a 4-point Likert scale, ranging from 1 (“rarely or none of the time”) to 4 (“most or all of the time”). The higher the score was, the higher the loneliness was. We chose this single-item measure of loneliness for two reasons: first, there is no longitudinal and nationwide survey of older Chinese which uses a standardized scale (i.e., ULCA). Second, different from specific research studies on the structure and differential experience of loneliness in the field of psychology, this study focused on the global perception of loneliness, and this single-item measure has been demonstrated to correlate highly with multi-item loneliness scales and has been widely used in previous studies [36,42,43,44].

#### 3.2.2. Social Participation

The item of social participation in the CHARLS questionnaire was selected to measure the social participation of the older adults: “Have you done any of these activities in the last month? ‘Interacted with friends’, ‘Played Ma-jong’, and other activities”. This study measured social participation from two dimensions: (1) Whether or not: According to existing research [45], social participation is encoded as a dummy variable, specifically, respondents who choose “none of these (12)” are encoded as “NO” and assigned a value of 0; the remaining 11 types of activities are encoded as “Yes” and assigned a value of 1. (2) Frequency: The CHARLS questionnaire contains the following items: “How often in the last month did/have you do?” Scores range from 1 (not regularly) to 3 (Almost daily); this is a continuous variable.

#### 3.2.3. Social Support

Social support was measured by the item “Suppose that in the future, you needed help with basic daily activities. Do you have relatives or friends (besides your spouse/partner) who would be willing and able to help you over a long period of time? What is the relationship to you of that person or those persons?”, which was well accepted by the Chinese older adults samples [34]. Six types of relatives were identified in the current study, such as parents, children and four other types. The number of types was taken as the measurement index of social support in the current study, and the range was 0–6, 0 indicating no social support.

#### 3.2.4. Covariates

The control variables were included in the analysis because of their well-documented associations with older adults’ loneliness or social participation [44,46,47]. These covariates included age; gender (female = 1); marital status (married = 1); education level (illiterate = 1; below primary school = 2; primary school graduate = 3; middle school and higher = 4); working status (working = 1); annual family income, which was log-transformed; self-rated health (poor = 1; fair = 2; good = 3); and living arrangement (alone = 0) (see Table 1).

### 3.3. Statistical Analyses

Hierarchical Linear Modeling (HML) was used as the major analytical framework in this study, which is more effective than traditional statistical analysis methods of longitudinal data (e.g., multiple repeated measure ANOVA). We adopted this model for the following reasons: it could be used to deal with incomplete or unbalanced panel data, and it allows the time interval to be different for every respondent. That is to say, this model makes it possible to analyze datasets that include respondents who did not participate in all survey waves [48].

Figure 1 depicts the conceptual and analytical framework of the current study. Paths a and b showed an indirect relationship between social participation and loneliness, mediated by social support. Path c’ reflected the direct effects of social participation on loneliness. The nature of longitudinal data is such that multiple observations of the same respondents are all correlated with each other. Social participation, social support, and loneliness were all at the observational level (level 1), nested within each individual (level 2).

We performed Hierarchical Linear Modeling (HML) to analyze the phenomenon of multiple observations nested within individuals from longitudinal studies. In order to test the first hypothesis, the unconditional average model and unconditional growth model were used to explore the patterns of loneliness trajectories among the older adults over time. Considering inter-individual heterogeneity, the year variable was estimated by using the fixed effect model, which is as follows:


**Unconditional average model (null model):**

(1)
$$\text{Level-1}:\;Y_{ij}=\beta_{0j}+e_{ij}\;$$


(2)
$$\text{Level-2}:\;\beta_{0j}=\gamma_{00}+\mu_{0j}$$



Substitute (2) into (1) to obtain the combined unconditional average model:(3)$$Y_{ij}=\gamma_{00}+\mu_{0j}+e_{ij}$$
where the dependent variable $Y_{ij}$ is the loneliness of individual *j* in the survey year *i*, $\beta_{0j}\;$ is the mean score of loneliness for individual *j*, $\gamma_{00}\;\;$is the intercept of the mean score of loneliness for each individual, and $e_{ij}\;$ and $\mu_{0j}\;\;$are random errors. Intra-class Correlation Coefficients (ICC) can be calculated according to the null model. ICC = inter-group variance/(inter-group variance + intra-group variance) = 31.6% (Table 2). Therefore, more than 30% of the overall variation in Y was caused by individual differences, and it is necessary to adopt HLM for estimation [49].

**Unconditional growth model**:(4)$$\text{Level-1}:\;Y_{ij}=\beta_{0j}+\beta_{1j}iyear_{ij}+e_{ij}$$(5)$$\text{Level-2}:\;\beta_{0j}=\gamma_{00}+\mu_{0j}$$(6)$$\beta_{0j}=\gamma_{00}+\mu_{0j}$$

Substitute (5) and (6) into (4) to obtain the combined unconditional growth model:$$Y_{ij}=\gamma_{00}+\left(\gamma_{10}+\mu_{1j}\right)iyear_{ij}+\mu_{0j}+e_{ijj}$$
where *iyear* is the survey year. Firstly, we used the unconditional growth model (Model 2) to test the loneliness trajectories of the older adults over time. Then, we analyzed the impact of social participation on loneliness, based on the baseline model, and then gradually included social participation (Model 3); frequency of social participation (Model 4) was included at level-1.

To answer the second research question, we conducted longitudinal mediation analysis [50], which makes it possible to examine the causal association between social participation, social support and loneliness [51], as well as reduce potential endogeneity [52]. Then, we performed a Bootstrap method to test the direct and indirect effects of social participation and the significance of the mediating effect, so as to explore whether social support had a mediating role.

## 4. Results

### 4.1. Descriptive Results

The descriptive results of all the variables in the analysis are presented in Table 3, including the repeated outcome measures. Due to the attrition of baseline samples and acquisition of new samples in this survey program, the distribution of demographic characteristics of the older adults in subsequent survey waves changed, with slight variation, such as gender and marital status distribution. Educational level rose over time, probably due to the inclusion of a new, younger sample of highly educated people in 2015 and/or 2018.

In addition, the average loneliness scores for the older adults were 1.58 (SD = 1.02, ranging from 1 to 4) during the whole survey period, which was generally at the middle and low level. With the increase in age, the level of loneliness increased from 1.47 to 1.66 (F = 77.270, *p* < *0*.001). The proportion of social participation that was yes at baseline was 55.07%, compared with 49.54% and 49.12% in 2015 and 2018, respectively. That is to say, the social participation of the older adults decreased year by year over time. Older adults’ access to social support slightly decreased, with the mean scores of total social support from 2013 to 2018 being 1.42, 1.38 and 1.35, respectively.

### 4.2. Loneliness Trajectory of the Older Adults

In general, the loneliness of the older adults from 2013 to 2018 was at a medium and low level, but there were age differences and cohort differences in the loneliness of the older adults. The average score of loneliness among the older adults from 2013 to 2018 was between 1.45 and 1.84, indicating that the loneliness level of the older adults was relatively low.

In this section, we re-categorized the birth year into seven consecutive cohorts including individuals born from 1910–1929 to 1950–1959 to examine the cohort trend in loneliness. Because the number of people born between 1910 and 1929 was rare (*n* = 324), those people were recorded as one group. Figure 2 showed the cohort trends in loneliness by age group. Overall, the average score of loneliness was higher in those born later than in those born earlier for each age group.

Figure 3 shows the age trends in loneliness by survey time point. For all three time points, the average scores of loneliness increased gradually with age. In 2013, the average score of loneliness among those aged 60–69 was 1.45 points, while in 2015 and 2018, the average score was 1.55 points and 1.62 points, respectively. Similarly, the average scores of loneliness among those 70–79 years old were 1.48, 1.65 and 1.71 points, respectively. A similar trend was found among those aged 80 or older. In order to further explore the differences in loneliness in the three survey points, we used the unconditional average model (Model 1) and the unconditional growth model (Model 2) to test the differences (Table 2). Model 1 indicated that the average score of loneliness among the older adults was 1.59 in the past 7 years. Model 2 showed the trajectory of loneliness from 2013 to 2018. It can be seen that the average score of loneliness is 1.58. Compared with 2013, loneliness increased in 2015, and the average score was 0.12 points higher. Loneliness in 2018 showed a significant increase, with the mean score being 0.21 points higher than that at baseline. The results of the HLM showed that the loneliness of the older adults had an upward trend from 2013 to 2018, thus verifying Hypothesis 1.1.

### 4.3. The Role of Social Participation on Loneliness Trajectories

Table 4 displays the HLM results about the effects of all three social participation indicators on loneliness. Model 3 examined the second research question. In each survey point, after controlling for potential confounding variables, social participation was significantly negatively correlated with loneliness, and the loneliness of older adults with social participation was 0.06 points lower than that of non-participation (S.E. = 0.013, *p* < 0.001). The mean loneliness in 2015 and 2018 was higher than the baseline, verifying Hypothesis 2.1. In addition, the loneliness of middle-old (β = −0.036, *p* < 0.01) and old-old (β = −0.065, *p* < 0.01) participants also increased with age. Loneliness was also higher among older adults who were unmarried, higher educated, had poor self-rated health, lived alone and had low income.

For the older adults with social participation, we further analyzed the influence of frequency (Model 4) on loneliness. Firstly, Model 4 presented the impact of frequency on loneliness, and the results showed that frequency had a significant negative impact on loneliness. With the increase in social participation, the loneliness of the older adults decreased. Hypothesis 2.2 was verified.

### 4.4. Path Analysis: The Mediating Role of Social Support

Firstly, we used HLMs to examine the direct impact of social participation on loneliness (Table 5). The results showed that social participation significantly negatively predicted loneliness, and the higher the frequency, the lower the loneliness. Then, we employed longitudinal mediation analysis [50] to test the mediating role of social support between social participation and loneliness (Table 5). Model 6 examined the direct relationship between social participation and social support. After controlling for covariates, social participation was associated with increased social support over time (β = −0.109, S.E. = 0.010, *p* < 0.001). Combining Model 5 with Model 7, we found that social participation still had a significant impact on loneliness after adding social support, but the regression coefficient reduced from −0.059 to −0.049, indicating that social support played a mediating role in the relationship between social participation and loneliness.

Secondly, we used a Bootstrap method to estimate the mediating role of social support, with covariates controlled. Table 6 shows the significance test results and we find that the mediation effect was statistically significant because the 95% confidence interval did not contain 0 [49]. Figure 4 and Table 6 show the mediating paths of social support. The results showed that the estimated indirect effect of social support was −0.011, and the 95% CI was [−0.013, −0.009], excluding 0, indicating that the mediation of social support was significant, and the indirect effect accounted for 18.3% of the total effect. Furthermore, the direct effect of social participation on loneliness was significant, with an estimated value of −0.049 (S.E. = 0.012) and a total effect of −0.060 (S.E. = 0.013). In other words, social participation can not only directly affect loneliness, but also indirectly affect loneliness through social support. Hypothesis 3.1 has been confirmed.

## 5. Discussion

Based on activity theory, this paper used HLM and longitudinal mediation analysis to explore the trajectory of loneliness among older adults in China, and examined the role of social participation and the mediating role of social support.

The results of the HLM unconditional average model and unconditional growth model indicated that from 2013 to 2018, the loneliness of Chinese older adults was at a medium level and showed an increasing trend, which is consistent with previous studies [6,53]. Meanwhile, there was also a cohort effect, which was different from some Western studies [54,55]. Hulur et al. (2016) pointed out that although the level of loneliness for older adults was higher than that of other age groups, it did not show an increasing trend and significant cohort difference, which may be related to the decreasing dependence one social environment that could trigger loneliness [55]. However, the dual attributes of age effect and cohort effect of loneliness among Chinese older adults may lie in unique social and cultural factors. It is not difficult to find that contemporary older adults experienced the period when Chinese Family Planning Policy was implemented most strictly (1980s and 1990s), leading to an increasing amount of nuclear families (i.e., only one child per family) and decreasing family size recently. Meanwhile, in the larger social context of accelerated urbanization, which increased from 53.73% in 2013 to 59.58% in 2018 [56], more and more young adults have migrated from home for employment, and physical distance between generations has also become much further. The traditional phenomena of “children and grandchildren round one’s lap” (*ersunraoxi*) and “four generations living together” (*sishitongtang*) are rare. In addition, the accelerated pace of life and the increase in work pressure have led to a simultaneous decline in the quantity and quality of intergenerational communication. Current Chinese society is still described as a “family orientation and filial piety culture”, where family and children are the core elements of emotional attachment and belonging for the older adults, while the reduction or absence of family functions directly increases the risk of loneliness for them [6,57].

Secondly, after verifying the time-varying characteristics of loneliness, this paper focused on the dynamic influence of social participation on loneliness. We found that after controlling for covariates, the increase in loneliness among older adults was associated with a decrease in social participation—that is, social participation negatively predicted loneliness, which was consistent with numerous existing cross-sectional studies [17,24,34]. We further analyzed those older adults who engaged in social participation, and explored the influence of frequency on loneliness. The results showed that frequency was negatively correlated with loneliness. Compared with the older adults who participated in social activities at a low frequency, the older adults with medium and high frequency had lower loneliness. Moreover, we also found that the proportion of older adults who engaged in social participation declined over time, as well as the frequency, which may be linked with rapid digitalization and construction since 2014. The data showed that the number of Chinese older internet users had increased from 11.7 million in 2013 to 54.7 million in 2018 [58]. Some studies reported that moderate internet usage had a positive effect on older adults’ mental health [59,60], while other studies indicated that frequent or problematic internet use could exacerbate loneliness through increasing social isolation, and reducing offline social interaction with family, friends or other people [61,62]. Considering the fact that older adults valued face-to-face social contacts more than online socializing and they were more vulnerable to problematic internet use or internet addiction [61], the slightly beneficial short-term effect of online activity may not be enough to compensate for the increased loneliness caused by reduced social participation in real life, which should be investigated in the future.

Thirdly, we examined the mediating role of social support. The results showed that there was a significant negative relationship between social support and loneliness, which was consistent with the previous conclusion [23,63]. As is known to all, older people would inevitably experience the dilemma of shrinking social network, loss of social value and reduced social status, which hinder older adults’ access to social support [22,23,31,32]. In order to alleviate negative emotions and consequences, they often obtained social support and social resources through social participation and building new social networks. Although socio-emotional selectivity theory pointed out that older adults may intentionally reduce unnecessary interaction and invest their limited energy in intimate relationships, Chinese older adults tended to put family members first, which has not been the case during the past 30 years [2,3,42]. The loss of close family relationships has made older adults redefine what the most valuable and intimate relationship is and shift focus to the network surrounding them for the sake of good later-life quality. The key to successful aging lies in balancing “aging loss” with “aging gain”, and one of the effective ways to achieve this is social participation [64]. Therefore, social participation not only through its direct effect alleviates the loneliness of older adults, but also can increase their social support.

In sum, the main contribution of this study is to analyze the temporal trend of loneliness among Chinese older adults by using representative longitudinal data, and to analyze the effect and path of social participation on loneliness. However, there are still limitations as follows: first, although longitudinal data can reduce the result bias caused by endogeneity problems to some extent, it cannot rule out the existence of other omitted factors that are not included in CHARLS. Second, the core variables (i.e., loneliness) were single-item scales rather than standardized scales, leading to the reliability of such measures possibly being questioned, considering that they are multi-dimensional concepts. However, given differences in the cultural and social environment in China, it may be difficult for Chinese older adults to understand the concept and connotation of loneliness used in Western culture and the single item may be adequate in this case [26]. Third, in addition to social support, there may be other related factors (i.e., subjective well-being [11,44], depression [1,37], personality traits [27]) moderating or mediating the relationship between social participation and loneliness, leading to the results of this paper possibly only reflecting part of the overall impact of social participation. Therefore, a database with more comprehensive indicators and finer dimensions is needed in the future to supplement the specific approaches for studying the relationship between social participation and loneliness, so as to provide a rigorous and rich theoretical framework and model overview. Finally, due to the limitations of data available, this paper did not take the effect of the COVID-19 pandemic into consideration. Some relevant research studies have shown that loneliness has become a global health concern caused by reduced social contact and activities due to enforced restrictions, such as lockdowns, self-quarantine, staying at home, and social distancing, particularly for older adults [65]. Given the ongoing catastrophic effect of the COVID-19 pandemic on global health, future work should further investigate and compare the relationship pattern among social participation, social support and loneliness before and after the COVID-19 pandemic.

Despite these limitations, our findings have both theoretical and practical implications. In terms of theoretical aspects, our results suggest that social participation is important not only because of its direct effect on loneliness, but also through increased social support in aiding older adults to alleviate loneliness. Additionally, with the further extension of aging and the continuous advancement of active aging processes in China, except for the rigid demands (e.g., health insurance), we should put older adults first and construct a living environment for aging construction, so as to encourage them to actively participate in various social activities, thus alleviating loneliness, improving life satisfaction and quality of life, and achieving the goal of healthy and successful aging.

## 6. Conclusions

This paper attempts to analyze the loneliness trajectory of Chinese older adults in recent years. The results show that the level of loneliness among them is on the rise, which provides empirical evidence for the government, society and public to value the psychological health problems caused by the extension of aging and formulate effective measures. Furthermore, by analyzing the effects and pathways of social participation on loneliness, we find that social participation can directly and indirectly alleviate loneliness, which also resonates with the active and healthy aging policy proposed by WHO.

## Figures and Tables

**Figure 1 ijerph-19-00815-f001:**
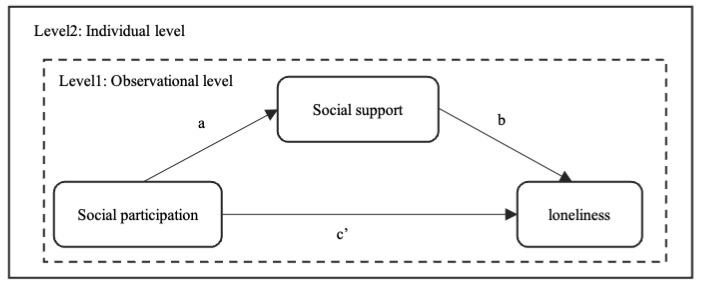
The core conceptual and analytical framework.

**Figure 2 ijerph-19-00815-f002:**
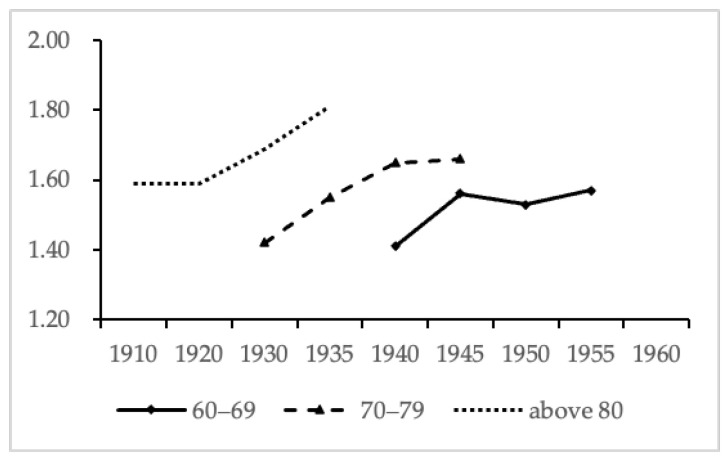
Cohort trends in loneliness by age group.

**Figure 3 ijerph-19-00815-f003:**
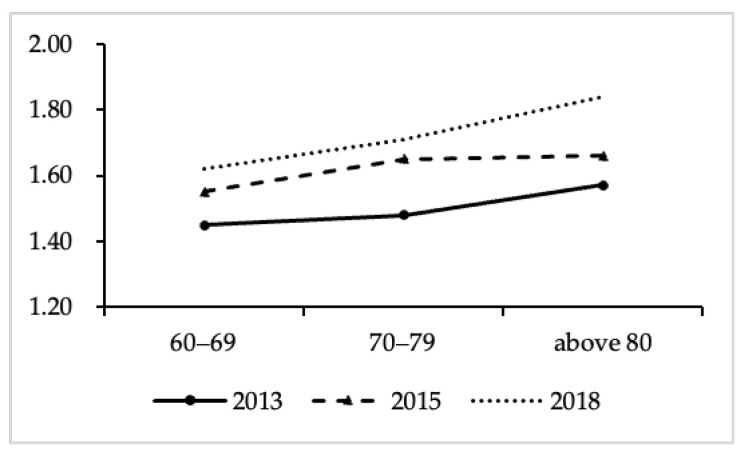
Age trends in loneliness by survey time point.

**Figure 4 ijerph-19-00815-f004:**
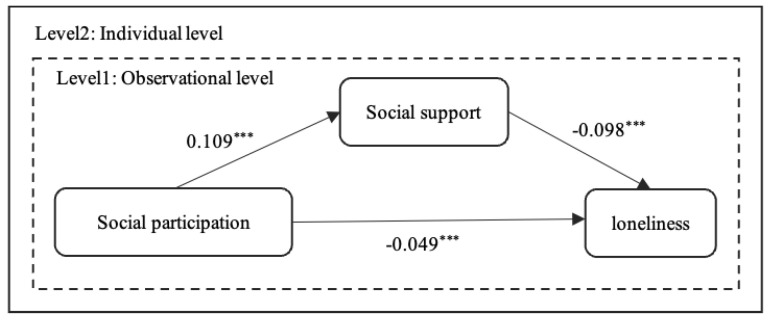
The influence of social participation on loneliness. *** *p* < 0.001.

**Table 1 ijerph-19-00815-t001:** Information about variables’ properties: number of items, response options, and coding procedure.

Variables	Description
Loneliness	How often in the last week did you feel lonely; 1–4, the higher the score, the higher the loneliness.
Social participation	Have you done any of these activities in the last month? 1—yes
Frequency of social participation	How often in the last month have you done them? 1–3, the higher the score, the higher the frequency
Social support	The total number of types of social support sources; range 0–6, the higher the number, the more social support they received.
Age period	1: 60–69-year-old, 2: 70–79, 3: 80 or older
Gender	0—male, 1—female
Marriage	0—non-married, 1—married (have partner)
Education level	1—illiterate, 2—below primary school, 3—primary school graduate, 4—middle school and higher
Work	0—not working, 1—working
Self-rated health	Self-rated health status; 1—poor, 2—fair, 3—good
Live arrangement	0—alone, 1—not alone

**Table 2 ijerph-19-00815-t002:** Hierarchical linear model of loneliness.

	Model 1		Model 2	
	Coefficient	S.E.	Coefficient	S.E.
**Fix effects**				
iyear (2013)				
2015			0.125 ***	0.014
2018			0.213 ***	0.014
intercept	1.582 ***	0.008	1.457 ***	0.012
**Random effects**				
Level 2 (individual)	0.330	0.010	0.337	0.010
Level 1 (observational)	0.715	0.009	0.702	0.009
**Observations**	25,192			
**Observation group**	12,232			
**ICC**	0.316		0.324	

Notes: (1) *** *p* < 0.001; (2) ICC = intraclass correlation coefficient.

**Table 3 ijerph-19-00815-t003:** Descriptive statistical characteristics of Chinese older adults.

Variables	M (SD)/%		
	2013	2015	2018
**N**	7208	8381	9603
**Loneliness**	1.47(0.92)	1.58(1.03)	1.66(1.07)
**Social participation (yes)**	55.02%	49.50%	49.12%
**Frequency of social participation**	2.12(0.81)	1.99(0.82)	2.03(0.81)
**Social support**	0.92(0.80)	0.91(0.80)	0.95(0.78)
**Age**	67.91(6.54)	68.03(6.52)	68.57(6.47)
**Gender (male)**	50.42%	49.52%	49.63%
**Marriage (married)**	80.31%	80.37%	80.13%
**Education level**			
illiterate	34.61%	33.72%	28.65%
below primary school	44.83%	44.39%	45.28%
primary school graduate	13.11%	13.88%	16.21%
middle school and higher	7.45%	8.02%	9.86%
Work (working)	54.38%	54.19%	53.11%
Self-rated health	1.91(0.72)	1.91(0.70)	1.91(0.71)
**Live arrangement (non-alone)**	100.00%	85.25%	85.08%
**Total household income (¥)**	25,254.07 (33,794.15)	25,018.83 (34,561.12)	34,523.39 (45,267.32)

**Table 4 ijerph-19-00815-t004:** Hierarchical linear modeling results of social participation and loneliness.

	Model 3: Social Participation on Loneliness		Model 4: Frequency of Participation on Loneliness	
	Coefficient	S.E.	Coefficient	S.E.
**Social participation**	−0.060 ***	0.013		
**Frequency of participation** (infrequent)				
Almost weekly			−0.059 **	0.020
Almost every day			−0.095 ***	0.020
**iyear (2013)**				
2015	0.091 ***	0.014	0.057 **	0.019
2018	0.187 ***	0.014	0.164 ***	0.019
**Age (60–69)**				
70–79	−0.036 **	0.015	−0.045 *	0.019
80 or older	−0.065 **	0.028	−0.015	0.037
**Gender (female)**	0.073 ***	0.015	0.062 **	0.019
**Marriage (married)**	−0.445 ***	0.020	−0.433 ***	0.025
**Education level (illiterate)**				
below primary school	−0.073 ***	0.016	−0.089 ***	0.022
primary school graduate	−0.148 ***	0.023	−0.160 ***	0.029
middle school and higher	−0.170 ***	0.027	−0.177 ***	0.033
**Work (working)**	0.040 *	0.013	0.064 **	0.018
**Self-rated health**	−0.241 ***	0.009	−0.230 ***	0.012
**Live arrangement (non-alone)**	−0.217 ***	0.023	−0.294 ***	0.031
**Total household income (log)**	−0.023 ***	0.003	−0.023 ***	0.004
**Intercept**				
level-2	2.758		2.808	
level-1	0.306		0.368	
**Observation**	25,192			
**Observation group**	12,232			
**ICC**	0.306		0.399	

Note: * *p* < 0.05, ** *p* < 0.01, *** *p* < 0.001.

**Table 5 ijerph-19-00815-t005:** Mediating pathway test results of the influence of social participation on loneliness.

	Model 5		Model 6		Model 7	
	Loneliness		Social Support		Loneliness	
	Coefficient	S.E.	Coefficient	S.E.	Coefficient	S.E.
**Social participation**	−0.059 ***	0.012	0.109 ***	0.010	−0.049 ***	0.012
**Social support**					−0.098 ***	0.008
**iyear (2013)**						
2015	0.093 ***	0.014	0.016	0.012	0.094 ***	0.014
2018	0.187 ***	0.014	0.047 **	0.012	0.191 ***	0.014
**Age (60–69)**						
70–79	−0.035 *	0.015	−0.007	0.012	−0.036 **	0.015
80 or older	−0.069 *	0.028	−0.008	0.022	−0.070 **	0.027
**Gender (female)**	0.076 ***	0.015	−0.003	0.012	0.076 ***	0.015
**Marriage (married)**	−0.444 ***	0.020	0.115 ***	0.016	−0.432 ***	0.020
**Education level (illiterate)**						
below primary school	−0.070 ***	0.016	0.022	0.013	−0.068 ***	0.016
primary school graduate	−0.145 ***	0.023	0.022	0.018	−0.143 ***	0.023
middle school and higher	−0.169 ***	0.028	−0.018	0.022	−0.167 ***	0.028
**Work (working)**	0.035 *	0.014	0.041 ***	0.011	0.040 *	0.014
**Self-rated health**	−0.244 ***	0.009	0.093 ***	0.007	−0.236 ***	0.009
**Live arrangement (non-alone)**	−0.212 ***	0.023	0.175 ***	0.019	−0.197 ***	0.023
**Total household income (log)**	−0.023 ***	0.003	0.013 **	0.003	−0.022 ***	0.003
**Intercept**						
Level-2	2.758		0.264		2.789	
level-1	0.472		0.294		0.463	
**Observation**	25,192					
**Observation group**	12,232					

Note: * *p* < 0.05; ** *p* < 0.01; *** *p* < 0.001.

**Table 6 ijerph-19-00815-t006:** Significance test results of mediating effect of social support.

	Coefficient of Paths			
	Path	Estimate	S.E.	95% CI
**direct effect**	c’	−0.049 ***	0.012	[−0.075, −0.023]
**Indirect effect**	ab	−0.011 ***	0.001	[−0.013, −0.009]
**Total effect**	ab + c’	−0.060 ***	0.013	[−0.086, −0.034]

Note: *** *p* < 0.001.

## Data Availability

We obtained the CHARLS data from the web http://charls.pku.edu.cn (accessed on 1 October 2021), which is available to users worldwide.

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
