# Peer review of "The Association between Social Participation and Loneliness of the Chinese Older Adults over Time—The Mediating Effect of Social Support"

_ijerph, 2022, doi:10.3390/ijerph19020815_

Round 1

Reviewer 1 Report

The study attempted to examine the longitudinal effect of social participation on loneliness and the mediating role of social support, as well as the cohort effect. While the study aimed to add to the “inadequate literature” and overcome the “shortcomings of previous empirical studies”, there is lacking references to the limitations of the existing studies. The manuscript will benefit from a clearer rationale for the research questions and an illustration of the importance, innovation and implication of the study.

Introduction

  • The authors are advised to use older people/adults instead of the elderly. The word choice associated with older people are inconsistent throughout the manuscript, e.g. elderly, older adults, please align them.
  • In lines 25 to 28, it says “Due to retirement, death of intimacy and migration of children, social environment that the elderly live has changed, and the quantity of the “empty-nesters” and “floating elderly” is gradually increasing, thus the prevalence of loneliness in older people is increasing.” Please provide the prevalence of loneliness in older people in China and some references on the associations between these factors and the increasing prevalence of loneliness.
  • Line 33: Why is it important to understand the characteristics and mechanism of loneliness?
  • In lines 63 to 65, it says “Continuity Theory believes that people only need to maintain their required level of social participation to obtain the optimal effect, which points out that the influence of social participation should be viewed dialectically”. What does “optimal effect” refer to? What does social participation “influence” that this relationship should be viewed dialectically?
  • From line 66, it says “For those who are active and extraverted, high frequency of social participation are conductive to lower loneliness, while for those who suffer from social dis orders or enjoy alone, an appropriate frequency of participation is also necessary.” Empirical evidence may be needed to demonstrate how individual characteristics influence their need for social participation in reducing loneliness. “An appropriate frequency of participation is also necessary” is unclear, do the authors mean that those who suffer from social dis orders or enjoy being alone do not need as much social participation as the active ones to keep their loneliness low? What does “social dis orders” mean? Also, how does this relate to the research questions?
  • In line 74, it says “…social participation is one of the important channels”. What does it mean? Does it mean social participation allows older people to expand their social network and thus more social support?
  • In line 85, it says “inadequate literature”, please explain with references.
  • In line 90, it says “In order to overcome the above shortcomings of previous empirical studies…”. Can the authors be more specific on the shortcomings of the previous studies please?
  • The introduction does not provide the rationale for examining the cohort effect (RQ 1).

Methods

  • The sample size is said to be N=25,192 (line 115). However, it says in line 159 that “The nature of longitudinal data is such that multiple observations of the same respondents are all correlated with each other”. Can the authors clarify what “sample size” means? Can the authors also provide the number of individual respondents, and how many of them have data from at least two time points?
  • In line 259, the authors may consider using the “young-old”, “middle-old”, and “old-old” instead of “middle elderly” and “senior elderly”.

Discussion

  • In line 335, it says “As is known to all, older people would inevitably experience the dilemma of shrinking social network, loss of social value and reduced social status, which hinder the elderly's access to social support. In order to alleviate negative emotions and consequences, they often obtained social support and social resources through social participation and building new social networks.” References are needed and there are theories (i.e. socioemotional selectivity theory) suggesting the otherwise that older people may not form new social networks.
  • What are the theoretical and clinical implications of the findings?

The authors are advised to proofread the manuscript thoroughly. Below are some examples but not an exhaustive list:

  • Line 54: …interventions that “increase” social participation…
  • Line 68: …those who suffer from social dis orders or enjoy “being alone”…
  • Line 113: “For the current analysis, the main respondents in 2013, 2015 and 2018, and aged 60 years or older.” The sentence does not make sense.
  • Line 314: “…population mobility fasts…” The sentence does not make sense.

Author Response

Response to Reviewer 1 comments (you can also find it in the attachment)

Thank you for your valuable and insightful comments. We have provided the responses to all your comments in two ways: (i) as for some individual comments, we response them one by one; (ii) as for other comments that related to one issue, we combined them and provided several unified answers.

Here my responses:

Introduction

Point 1: The authors are advised to use older people/adults instead of the elderly. The word choice associated with older people are inconsistent throughout the manuscript, e.g. elderly, older adults, please align them.

Response1(hereinafter referred to as R*): Thank you for your valuable advice. And we have aligned the term word by “older adults” in the whole article. A few examples are as follows:

Page 1 lines 2-3: Title: The Association between Social Participation and Loneliness of the Chinese Older Adults over time - the Mediating Effect of Social Support

Page 1 lines 9-12: Abstract: Based on activity theory, … in Chinese Older Adults, as well as the mechanism of them

…..

Point 2: In lines 25 to 28, it says “Due to retirement, death of intimacy and migration of children, social environment that the elderly live has changed, and the quantity of the “empty-nesters” and “floating elderly” is gradually increasing, thus the prevalence of loneliness in older people is increasing.” Please provide the prevalence of loneliness in older people in China and some references on the associations between these factors and the increasing prevalence of loneliness.

Response 2: Thank you for your constructive suggestion, and we have readjusted the statement shown as follows:

Page 1 lines 25-33: It’s important to note that China is also experiencing a rapidly developing economy and fast urbanization, both of which have drove young adults’ migration from home and distributed the traditional family structure, resulting in lager number of the non-traditional older adults population, such as “empty-nester” [1,2] and “floating older adults” [3]. All the above social changes contribute to increase the vulnerability of older adults to the experience of loneliness. The China National Committee on Ageing reported that nearly a third of the young-old adults felt lonely, and the prevalence of that in old-old individuals was higher than 50% in China in 2018 [4], and the proportion of empty-nest group who felt lonely was even 78.1% [1].

The references are as follow:

1] Wang, G., Hu, M., Xiao, S. Y., Zhou, L. Loneliness and depression among rural empty-nest elderly adults in Liuyang, China: a cross-sectional study. Bmj Open 2017, 7(10), e016091. doi: 10.1136/bmjopen-2017-016091.

2] Li, Y. W., Zhou, L. S. Research on the lonely situation and intervention strategy of empty-nest elderly. Chinese Journal of Gerontology 2016, 36(11), 2809–2811. doi:CNKI:SUN:ZLXZ.0.2016-11-119.

3] Cao, G., Nie, Q. A study on the Social Adaptation of the Old Floating Race from the Perspective of Social Integration. Future and Development 2017, 11, 66-69+58. doi: CNKI:SUN:WLYF.0.2017-11-013.

4] Dang, J., Li, J., Zhang, Q., Luo, X. Development Report on The Quality of Life for The Elderly in China (2019). Beijing: Social Science Academic Press 2019: 5-28, 146-161.

Point 3: Line 33: Why is it important to understand the characteristics and mechanism of loneliness?

Response 3: Thank you for your question. We are sorry that this statement made you confused, and here we want to highlight the importance of understanding the characteristics and mechanism of loneliness from a longitudinal perspective. Then, we have revised and supplemented this section and the detail content are as follow:

Page 1 lines 36-53: Loneliness is a psychological experience that changes constantly over time. Several cross-temporal meta-analysis literatures investigated changes in Chinese older adults’ loneliness through correlating loneliness scores with several social indicators that including urbanization level, divorce rate, unemployment rate, and revealed a birth cohort increase in loneliness levels in Chinese older adults [6, 9], that’s to say the level of loneliness among more recent birth cohorts of Chinese older adults was higher than those earlier birth cohorts. In parallel, evidences from systematic reviews and empirical re-searches indicate that loneliness can be prevented or relieved by interventions such as increased social engagement and social contacts. Taken together with the serious situation that both prevalence of loneliness among older Chinese nowadays and proportion of empty-nest, floating elderly increases continuously in the future, it’s vital and valuable to have a comprehensive understanding of temporal nature and mechanism of loneliness among older adults in China, a culture that emphasize older first and filial piety. There exist abundant theoretical and empirical studies focusing on the fac-tors-related and consequences of loneliness in terms of cross-sectional perspective [6,8,9], but the study on the characteristics (e.g., time-varying) of itself and mechanism (e.g., mediator and moderator) based on a longitudinal and large-scale data is not sufficient, especially for the Chinese older adults in recent 10 years, a period that happened dramatic social change [13].

Point 4: In lines 63 to 65, it says “Continuity Theory believes that people only need to maintain their required level of social participation to obtain the optimal effect, which points out that the influence of social participation should be viewed dialectically”. What does “optimal effect” refer to? What does social participation “influence” that this relationship should be viewed dialectically?

Response 4: Thank you for this question. We are sorry that this section made you confused, and here we want to provide an opinion that the impact of social participation on loneliness varies with different participation frequency and whether individuals are autonomic or obligative. Then, we have revised this section and the detail content are as follow:

           Page 2 lines 84-91: In addition, Continuity Theory believes that people only need to maintain their required frequency of social participation to obtain the optimal effect on their physical and mental health, which points out that the influence of social participation should be viewed dialectically [14] and highlights individuals’ autonomy and appropriateness of frequency [25,26]. A study examined the cross-sectional associations of the type, frequency and autonomy for social participation with physical and mental health, and the results showed that only autonomic and appropriately frequent (i.e., weekly, monthly rather than daily or yearly) social participation had positive impact on mental health [25].

Point 5: From line 66, it says “For those who are active and extraverted, high frequency of social participation are conductive to lower loneliness, while for those who suffer from social dis orders or enjoy alone, an appropriate frequency of participation is also necessary.” Empirical evidence may be needed to demonstrate how individual characteristics influence their need for social participation in reducing loneliness. “An appropriate frequency of participation is also necessary” is unclear, do the authors mean that those who suffer from social dis orders or enjoy being alone do not need as much social participation as the active ones to keep their loneliness low? What does “social disorders” mean? Also, how does this relate to the research questions?

Response 5: Thank you for your comments and advice. First, we have revised the statements and add the references-related. On your second and third questions, what we are trying to say is that the impact of social participation on loneliness varies with individual differences, which is agreement with your understanding. As for the last one, we propose an inference (the last sentence in this paragraph) based on the above opinion. Below are the specific contents:

Page 2-3 lines 92-100: The best effect of participation type and frequency on loneliness may be related to individual characteristics such as age, socio-economic status, and personality traits. Former studies relevant to personality traits in older adults reported that neuroticism and extraversion were the most influential personality factors [27]. Specifically, for those who were active and extraverted, high frequency of social participation are conductive to lower loneliness, while for those who suffered from social disorders (i.e., anxiety about talking to others in public) or enjoy being alone, an infrequent participation may be better. In sum, it isn’t difficult to infer that the influence of social participation on loneliness also has differences in participation intention and frequency.

Point 6: In line 74, it says “…social participation is one of the important channels”. What does it mean? Does it mean social participation allows older people to expand their social network and thus more social support?

Response 6: Thank you for your questions. As you understand, the older adults tend to expand their social network they have lost as aging through social participation, which has been demonstrated in many studies. And the detailed contents are as follows:

Page 3 Lines 111-122: Within the context of traditional family structure and collective culture in China, older adults tend to put the family member first, especially children, then friends and peers [23]. However, the last 30 years have seen drastic declines in fertility, diluted filial piety, and uneven rates of economic mobility, all of which have contributed to rapid increases in empty-nest households and in the proportions of left-behind older adults whose adult children leave home for employment, more family fragmentation and smaller family size prevent older adults from receiving family support, thus, they fail to nurture their desired family relationship. In order to compensate for the loss of close family attachment and to alleviate the potential loneliness, they attempt to seek alternative sources of support. Hence social participation become one of the important channels for them to substitute family network and a convoy of late-life social new network [30].

Point 7: In line 85, it says “inadequate literature”, please explain with references.

Point 8: In line 90, it says “In order to overcome the above shortcomings of previous empirical studies…”. Can the authors be more specific on the shortcomings of the previous studies please?

Responses 7-8: Thank you for these two questions and what we express here is about research status and potential limitations in this field. Then, we combine these two questions for a unified answer and show it below:

Page 3 lines 140-151: In a nutshell, although the direct effects of social participation on loneliness or social support on loneliness among older adults have widely explored at home and abroad, very few studies tested this association using longitudinal and large-scale nationwide samples, which is necessary to draw conclusions on the casual relationship between variables and also called up in some former studies [40,41]. Although systematic reviews of interventional studies show that trainings on both social participation and social support had beneficial effects on loneliness among older adults [10-12] which address the casual relationships between variables, to our knowledge, there is no relevant research that regarding social participation as an interventional strategy to alleviate loneliness among older Chinese. Besides, interventional studies showed several limitations, such as small and convenience samples, lack of considerations about the dynamic traits of loneliness according to life circumstances and aging.

Point 9: The introduction does not provide the rationale for examining the cohort effect (RQ 1).

Response 9: Thank you for this valuable comment. The research question 1 was raised for consideration about dramatic social changes in recent 10 years in China, meanwhile several studies addressed a birth cohort increase in longlines levels in Chinese older adults from 1995 to 2011, which employed a special statistic method -- the cross-temporal meta-analysis, which makes it possible to study birth cohort differences in scores on loneliness. We have supplemented the evidence in the introduction and the more detailed content can also be present as follow:

Page 1 lines 36-42: Two cross-temporal meta-analysis literatures investigated changes in Chinese older adults’ loneliness through correlating loneliness scores with several social indicators that including urbanization level, divorce rate, unemployment rate, and revealed a birth cohort increase in loneliness levels in Chinese older adults [6, 9], that’s to say the level of loneliness among more recent birth cohorts of Chinese older adults was higher than those earlier birth cohorts.

Methods

Point 10: The sample size is said to be N=25,192 (line 115). However, it says in line 159 that “The nature of longitudinal data is such that multiple observations of the same respondents are all correlated with each other”. Can the authors clarify what “sample size” means? Can the authors also provide the number of individual respondents, and how many of them have data from at least two time points?

Response 10: Thank you for your comments and suggestions. We use data from the China Health And Retirement Longitudinal Studies, a public nationwide survey performed by Peking University

Page 4 lines 193-197: our final sample size was 25192 which including 7208, 8381, and 9063 respondents in 2013, 2015, and 2018, respectively. There were 12232 individual respondents, and the proportions of respondents who participated in one wave, two waves, and three waves were 30%, 33%, and 37%, respectively.

Point 11: In line 259, the authors may consider using the “young-old”, “middle-old”, and “old-old” instead of “middle elderly” and “senior elderly”.

Response 11: Thank you for your valuable suggestion. We have revised these terms and you can find them in the following paragraph:

Page 10 lines 362-364: The mean loneliness in 2015 and 2018 higher than the baseline respectively, verifying hypothesis 2.1. In addition, loneliness of middle-old (β = -0.036, p <0.01) and old-old (β = -0.065, p <0.01) also increased with age.

Discussion

Point 12: In line 335, it says “As is known to all, older people would inevitably experience the dilemma of shrinking social network, loss of social value and reduced social status, which hinder the elderly's access to social support. In order to alleviate negative emotions and consequences, they often obtained social support and social resources through social participation and building new social networks.” References are needed and there are theories (i.e. socioemotional selectivity theory) suggesting the otherwise that older people may not form new social networks.

Response 12: Thank you very much for your valuable suggestions. Accordingly, we have revised the section and you can find it below:

Page 13 lines 451-461: As is known to all, older people would inevitably experience the dilemma of shrinking social network, loss of social value and reduced social status, which hinder the older adults' access to social support [22,23,31,32]. In order to alleviate negative emotions and consequences, they often obtained social support and social resources through social participation and building new social networks. Although Socioemotional Selectivity Theory pointed out that older adults may intentionally reduce unnecessary interaction and invest their limited energy in intimate relationships, Chinese older adults tended to put family memberships first which have not been met during the past 30 years [2,3,43]. The loss of close family relationships made older adults redefine what the most valuable and intimate relationship was and shift focus to the network surrounding them for the sake of good later-life quality.

Point 13: What are the theoretical and clinical implications of the findings?

Response 13: Thank you for this comment. We are sorry that the part about implications made you confused. Actually, we explicated the implications that you can find in the end of this conclusions. In order to make the implications much clearer, we have relocated them in the end of discussion.

Page 14 lines 492-500: Despite of the limitations, our findings have both theoretical and practical implications. In terms of theoretical aspect, our results suggest that social participation is important not only because of its direct effect on loneliness but also through increased social support in aiding older adults to alleviate loneliness. Additionally, with the further deepening of aging and the continuous advancement of active aging process in China, except for the rigid remands (e.g., health insurance), we should put the older adults first and construct the living environment for aging construction, so as to stimulate them to actively participate in various social activities, thus to alleviate loneliness, improve life satisfaction and quality of life, and achieve the goal of healthy and successful aging.

Point 14: The authors are advised to proofread the manuscript thoroughly. Below are some examples but not an exhaustive list:

  • Line 54: …interventions that “increase” social participation…
  • Line 68: …those who suffer from social dis orders or enjoy “being alone”…
  • Line 113: “For the current analysis, the main respondents in 2013, 2015 and 2018, and aged 60 years or older.” The sentence does not make sense.
  • Line 314: “…population mobility fasts…” The sentence does not make sense.

Response 14: Thank you for your insightful suggestions. In order to more accurately express the content, we proofread the manuscript thoroughly and revised the ambiguous and unacademic statements as possible as we could. Below are some examples and more detailed revision could be seen the red font in the revised manuscript:

Page 2 lines 75-77: Similarly, … that increasing social participation …

Page 2 Line 98: or enjoy being alone, an infrequent participation may be better.

Page 3 lines 135-136: Given the relatively higher and continuously increasing prevalence of Chinese older adults’ loneliness, …

Page 4 Line 193-195: our final sample size was 25192 which including 7208, 8381, and 9063 respondents in 2013, 2015, and 2018, respectively.

Page 12 Line 417-419: more and more young adults have migrated from home for employment and physical distance between generations also has become much longer.

Reviewer 2 Report

I deeply appreciate the opportunity to review the manuscript titled "The Association between Social Participation and Loneliness of 2 the Chinese Elderly over time - the Mediating Effect of Social 3 Support". Through a longitudinal study, the authors aim to investigate the temporal characteristics of loneliness and the influence of social participation in Chinese older adults.

Overall, the article is well structured, but does require substantial language revisions by an English native to improve its clarity (e.g., were lower likely to be lonely). It was difficult to follow the authors’ ideas at times, and I strongly advise a second review round after major textual revisions are performed. Below are some comments that I would advise the authors to analyze.

[Introduction]

Page 1, line 25 – The sentence “Due to retirement, death of intimacy and migration of children, social environment that the elderly live has changed (…)” is rather confusing to the reader. Please revise.

Page 1 (and overall) – Please revise the term “elderly” to “older adults” throughout the manuscript, as the first term is no longer well-received in geriatric/gerontology publications.

Page 2, lines 58-59 – What do you mean by “Unlike the accumulation and 58 enrichment of social participation research on loneliness of the Western elderly”?

Page 2, lines 63-65 – Please revise the sentence that starts with “In addition, Continuity Theory believes that people only need to maintain their (…)” as it is extremely confusing at it is.

Page 2, line 81 – What do you mean by “Given the relatively disadvantaged mental health status of Chinese elderly”?

Page 2, line 85-87 – Please consider removing the sentence that starts with “The current study also adds to the inadequate literature in general (…)”. Such statements would be necessary for a discussion/conclusion section.

[Data and Method]

Page 3, line 108 – Please revise “The resource of data” to “Data source”

Page 3, lines 113-114 – Please revise the sentence “For the current analysis, the main respondents in 2013, 2015 and 2018, and aged 60 years or older”. Moreover, please remove the “(N=25192)” as it is just a repetition of the information provided before.

Page 3, line 134 – “According to Tao and Shen (2014)”, please revise the citation style as per MDPI rules.

Concerning the Methods section, several omissions must be written in the manuscript, for example:

  1. The authors fail to indicate the study design;
  2. How did the authors have access to data from CHALRS?
  • Eligibility criteria seem rather vague;
  1. The authors fail to address any efforts made to address potential sources of bias
  2. Did the authors do not explain how missing data were addressed;
  3. How did the authors deal with the loss to follow-up between 2013, 2015 and 2018?
  • No ethical considerations are provided in this section.

[Discussion]

Page 10, lines 300-302 – Consider removing this specific segment “The main conclusions were 300 as follows. First of all”. Start the following sentence with “The results of HLM (…)”.

The authors should address their initial questions and hypothesis when discussing their findings. A deeper analysis of the significance of their findings in China’s current socio-political and cultural landscape would considerably improve the manuscript.

[Conclusions]

Page 11, line 364 – Please consider removing the sentences after “With the further deepening of ageing (…)” since it is not a conclusion of your work.

Author Response

Response to Reviewer 2 Comments (you can also find it in the attachment)

Thank you for your valuable and insightful comments. We have provided the responses to all your comments one by one.

Here my responses:

[Introduction]

Point 1: Page 1, line 25 – The sentence “Due to retirement, death of intimacy and migration of children, social environment that the elderly live has changed (…)” is rather confusing to the reader. Please revise.

Response1: Thank you for your valuable suggestion. We are sorry for that this statement made you confused, and we have revised it in the revised manuscript. More detailed information could be shown as follows:

Page 1 lines 25-30: It’s important to note that China is also experiencing a rapidly developing economy and fast urbanization, both of which have drove young adults’ migration from home and distributed the traditional family structure, resulting in lager number of the non-traditional older adults population, such as “empty-nester” [1,2] and “floating older adults” [3]. All the above social changes contribute to increase the vulnerability of older adults to the experience of loneliness.

Point 2: Page 1 (and overall) – Please revise the term “elderly” to “older adults” throughout the manuscript, as the first term is no longer well-received in geriatric/gerontology publications.

Response 2: Thank you for your valuable advice. And we have aligned the term word by “older adults” in the whole article. A few examples are as follows:

Page 1 lines 2-3: Title: The Association between Social Participation and Loneliness of the Chinese Older Adults over time - the Mediating Effect of Social Support

Page 1 lines 9-12: … explore the temporal variation characteristics of loneliness and the influence of social participation on loneliness in Chinese Older adults, as well as the mechanism of them

……

Point 3: Page 2, lines 58-59 – What do you mean by “Unlike the accumulation and enrichment of social participation research on loneliness of the Western elderly”?

Response 3: Thank you for your comment. We are sorry for that this statement made you confused and what we say here is that the quantity of relevant studies based on Chinese older adults was smaller than that within the context of Western culture. Then, we have revised the sentence as follow:

Page 2 lines 79-82: Despite of the existing academic researches of the association between social participation and loneliness among the Western older adults, the relevant studies based on Chinese older adults are relatively scarce …

Point 4: Page 2, lines 63-65 – Please revise the sentence that starts with “In addition, Continuity Theory believes that people only need to maintain their (…)” as it is extremely confusing at it is.

Response 4: Thank you for your comment. We are sorry for that this statement made you confused and have revised the sentence:

Page 2 lines 84-88: In addition, Continuity Theory proposes that people only need to maintain their required frequency of social participation to obtain the optimal effect on their physical and mental health, which points out that the influence of social participation should be viewed dialectically [14] and highlights individuals’ autonomy and moderate frequency rather than the more the better.

Point 5: Page 2, line 81 – What do you mean by “Given the relatively disadvantaged mental health status of Chinese elderly”?

Response 5: Thank you for your comment. We are sorry for that this statement made you confused and we have readjusted the sentence in order to make a clearer meaning.

Page 3 lines 135-136: Given the relatively higher and continuously increasing prevalence of Chinese older adults’ loneliness …

Point 6: Page 2, line 85-87 – Please consider removing the sentence that starts with “The current study also adds to the inadequate literature in general (…)”. Such statements would be necessary for a discussion/conclusion section.

Response 6: Thank you for your valuable suggestion and we have deleted and revised this section in the revised manuscript (page 3).

[Data and Method]

Point 7: Page 3, line 108 – Please revise “The resource of data” to “Data source”

Response 7: Thank you for this suggestion and we have revised this statement in the revised manuscript (p).

Page 4 line 177: 3.1 Data Source

Point 8: Page 3, lines 113-114 – Please revise the sentence “For the current analysis, the main respondents in 2013, 2015 and 2018, and aged 60 years or older”. Moreover, please remove the “(N=25192)” as it is just a repetition of the information provided before.

Response 8: Thank you for this suggestion that is helpful for us and we have revised this section in the revised manuscript.

Page 4 lines 184-197: For the current analysis, the main respondents in 2013 aged 60 years or older were first selected (N1=8934) ... Then, after further deleting some missing values of key independent variables and covariates, our final sample size was 25192 which including 7208, 8381, and 9063 respondents in 2013, 2015, and 2018, respectively.

Point 9: Page 3, line 134 – “According to Tao and Shen (2014)”, please revise the citation style as per MDPI rules.

Response 9: Thank you for this suggestion that is helpful for us and we have revised this citation style in the revised manuscript (see, page 5 lines 227-231).

Point 10: Concerning the Methods section, several omissions must be written in the manuscript, for example:

The authors fail to indicate the study design;

How did the authors have access to data from CHALRS?

Eligibility criteria seem rather vague;

The authors fail to address any efforts made to address potential sources of bias

Did the authors do not explain how missing data were addressed;

How did the authors deal with the loss to follow-up between 2013, 2015 and 2018?

No ethical considerations are provided in this section.

Response 10: Thank you for your insightful comments. Firstly, as for the study design, you can find it in the section of 3.3 statistical analysis, as well as the reasons for the way we deal with the missing data. Secondly, we displayed the way to obtain the data and the ethical considerations in the [Data Availability Statement] and [Institutional Review Board Statement] as following the text, respectively. Then, the remaining questions are all about the Methods, and we combine them for a unified answer. Below are the detailed information:

Page 4 lines 177-201:

3.1 Data Source

The data for this study come from three waves of the Chinese Health and Retirement Longitudinal Studies (CHALRS) conducted by the Institute of Social Science Survey (ISSS) of Peking University. The CHARLS adopted a multistage stratified probability-proportionate-to-size design to collect a nationally representative sample of Chinese residents aged 45 years and older. We obtain the data from the official web http://charls.pku.edu.cn , which is available to users worldwide.

For the current analysis, the main respondents in 2013 aged 60 years or older were first selected (N1=8934). The standard excluded from final analysis of data was data missing on the key variables (i.e., the loneliness measure). First, if a respondent has missing item on loneliness measure, then she/he is excluded from the analysis. Approximately 13% of those excluded from the analysis were missing on the loneliness measure. According to this standard, 1158, 963, and 1426 respondents in 2013, 2015 and 2018, respectively, were excluded from the analysis for this reason, and they are not significantly different from those kept in the analysis in terms of their key demographic information, including gender, age, income, marriage status and education. Then, after further deleting some missing values of key independent variables and covariates, our final sample size was 25192 which including 7208, 8381, and 9063 respondents in 2013, 2015, and 2018, respectively. There were 12232 independent respondents, and the proportions of respondents who participated in one wave, two waves, and three waves were 30%, 33%, and 37%, respectively.

Ethical approval for collecting data on human subjects was received at Peking University by their institutional review board.

Page 6 lines 246-253:

3.3 Statistical Analysis

Hierarchical Liner Model (HML) ... We adopted this model for the following reasons: first, it could be used to deal with incomplete or unbalanced panel data, and it allows the time interval different for every respondent. That’s to say, this model makes it possible to analyze dataset that included respondents who didn’t participate in all survey waves [49].

[Discussion]

Point 11: Page 10, lines 300-302 – Consider removing this specific segment “The main conclusions were as follows. First of all”. Start the following sentence with “The results of HLM (…)”.

Response 11: Thank you for this suggestion that is helpful for us and we have deleted this section in the revised manuscript.

Page 12 lines 403-404: …and examined the role of social participation and the mediating role of social support.

The results of HLM unconditional average model …

Point 12: The authors should address their initial questions and hypothesis when discussing their findings. A deeper analysis of the significance of their findings in China’s current socio-political and cultural landscape would considerably improve the manuscript.

Response 12: Thank you for your insightful suggestions. We have supplemented and readjusted the content in the Discussions section, and below are some examples:

Page 12 Lines 416-426: Meanwhile, in the larger social context of accelerated urbanization, more and more young adults have migrated from home for employment and physical distance between generations also has become much longer. The traditional phenomenon of “children and grandchildren round one's lap” (ersunraoxi) and “four generations living together” (sishitongtang) is rare. In addition, the accelerated pace of life and the increase of work pressure have led to the simultaneous decline in the quantity and quality of intergenerational communication. Current Chinese Society is still described as an “family orientation and filial piety culture”, where family and children are the core elements of emotional attachment and belonging for the older adults, while the reduction or absence of family functions directly increases the risk of loneliness for them [58].

Page 13 Lines 458-461: Chinese older adults tended to put family memberships first which have not been met during the past 30 years [2,3,43]. The loss of close family relationships made older adults redefine what the most valuable and intimate relationship was and shift focus to the network surrounding them for the sake of good later-life quality.

[Conclusions]

Point 13: Page 11, line 364 – Please consider removing the sentences after “With the further deepening of ageing (…)” since it is not a conclusion of your work.

Response 13: Thank you for your pertinent suggestion. We have deleted this section from the conclusions following your advice and you can find it in the revised manuscript (see page 14 Conclusion section).

Reviewer 3 Report

The 2013, 2015 and 2018 waves of a Chinese longitudinal survey were analysed to investigate changes in time in loneliness and the relationship with social participation and social support. The study found increased loneliness over time and the partial mediating role of social support between social participation and loneliness.

Here my comments:

  • Page 1, 32-33: “but the 32 discussion on the characteristics (e.g., time-varying) of itself and mechanism (e.g., media- 33 tor and moderator) are not sufficient, especially for the Chinese elderly”. I found this sentence vague; please consider reformulating to help the reader in understanding what you mean by characteristics and mechanism.
  • Socioemotional selectivity theory (Carstensen) is here missing. The authors should cite the theory in the introduction and then use it to discuss their results.
  • In the introduction, I would add a brief description of the relationship between loneliness and social participation and the individual variables you used as covariates (the ones you mentioned in rows 144-147 and present in Table 1)
  • Please add the rationale for considering social support as a mediator between social participation and loneliness and not: social participation as a mediator between social support and loneliness or loneliness as a mediator between social participation and social support or loneliness as a mediator between social support and social participation
  • Throughout the paper. Term “influence”. This reminds me of causality, while here no conclusion can be made on causality. I would prefer terms as “relate”.
  • Row 115. I would only report once the numerosity
  • Table 1 does not report any descriptive statistics, please revise its caption.
  • I would add what you think the covid-19 pandemic may have changed. Do you think loneliness is greatly increased and social participation changed? The present results continue to be informative?
  • I would also discuss the possibility of deep investigation of the relationship between social participation, social support, and loneliness considering other individual characteristics, e.g. personality traits.
  • I would also mention more explicitly some limitations of the present study. For instance, I suggest considering that you did not have any measure of wellbeing. This lack should be discussed. What if increased loneliness did not reflect decreased wellbeing? Maybe this is not the picture, based on literature, but a discussion of this could be added.

Author Response

Response to Reviewer 3 Comments (you can find it in the attachment)

Thank you for your valuable and insightful comments. We have provided the responses to all your comments one by one.

Here my responses:

Point 1: Page 1, 32-33: “but the discussion on the characteristics (e.g., time-varying) of itself and mechanism (e.g., mediator and moderator) are not sufficient, especially for the Chinese elderly”. I found this sentence vague; please consider reformulating to help the reader in understanding what you mean by characteristics and mechanism.

Response 1: Thank you for your comment. We are sorry that this statement made you confused and we have revised it as follow:

    Page 2 lines 50-53: … but little research based on a longitudinal and large-scale data has considered time-varying characteristics of older adults’ loneliness and predictors or determining factors, especially for the Chinese older adults in recent 10 years, a period that is happening dramatic social changes.

Point 2: Socioemotional selectivity theory (Carstensen) is here missing. The authors should cite the theory in the introduction and then use it to discuss their results.

Response 2: Thank you for your valuable suggestion. We have supplemented the Socioemotional selectivity theory in the Introduction and Discussion sections. Below is part of revised content:

[Introduction]

    Page 3 lines 107-113: In contrast, Social-Emotional Selectivity Theory points out that unlike the youngers expanding their social network through actively engaging in various social participation, older adults tend to intentionally shrink their network and select the most valuable and intimate relationships to invest in [29], which may be not appliable to Chinese older adults in current society. Within the context of traditional family structure and collective culture in China, older adults tend to put the family member first, especially children, then friends and peers [23]. …

[Discussion]

    Page 13 lines 456-461: Although socio-emotional selectivity theory pointed out that older adults may intentionally reduce unnecessary interaction and invest their limited energy in intimate relationships, Chinese older adults tended to put family memberships first which have not been met during the past 30 years [2,3,43]. The loss of close family relationships made older adults redefine what the most valuable and intimate relationship was and shift focus to the network surrounding them for the sake of good later-life quality. …

Point 3: In the introduction, I would add a brief description of the relationship between loneliness and social participation and the individual variables you used as covariates (the ones you mentioned in rows 144-147 and present in Table 1)

Response 3: Thank you for your valuable suggestions, and we have supplemented the section:

Page 2 Lines 68-70: Social participation not only can provide social relationships and resources to make up for their loss [16], but also build new social network to expand their existing ones, which are prerequisite for lower loneliness.

Page 3 Lines 129-135: In addition to social participation pattern (i.e., frequency [25]) and social support sources (i.e., children vs social network members [23,30]), the association among social participation, social support and loneliness may vary with individual differences, including age, gender, marital status, living arrangement, health status, and economic conditions [1,3,8,9]. For example, some previous studies showed that compared with those older adults who lived with children, the empty-nesters older adults were more likely to benefit from various social activities [1].

Point 4: Please add the rationale for considering social support as a mediator between social participation and loneliness and not: social participation as a mediator between social support and loneliness or loneliness as a mediator between social participation and social support or loneliness as a mediator between social support and social participation

Response 4: Thank you for this prominent comment. We think what you want to say and concern about is the bi-directional causality or the potential endogeneity among these variables. We will answer this question from the following two aspects [Note: the below explanation was not shown in the manuscript]:

Firstly, in terms of theoretical aspect, the previous studies proposed that social participation and social support were predictors of loneliness and social participation was a source of social support. Additionally, the existing studies demonstrated the mediating role of social support between social participation and mental health. In sum, we inferred that social support was a mediator between social participation and loneliness.

Secondly, this paper adopted a longitudinal data and longitudinal mediation model analysis method to address this causality. Bidirectional causality or endogeneity may be caused by the simultaneity of loneliness, social participation and social support, or other variables that have been neglected, and an effective way to solve this problem was to use large longitudinal datasets including multiple time points [53]. More seriously, the longitudinal mediation model was another effective way to demonstrate the direction of causality [54].

Point 5: Throughout the paper. Term “influence”. This reminds me of causality, while here no conclusion can be made on causality. I would prefer terms as “relate”.

Response 5: Thank you for this comment, as shown in R4, this paper could demonstrate the causality, so that we think the terms “influence” rational.

Point 6: Row 115. I would only report once the numerosity

Response 6: Thank you for your valuable suggestion and we have revised this statement in the revised manuscript.

Page 4 lines 193-194: after further deleting some missing values of key independent variables and covariates, our final sample size was 25192.

Point 7: Table 1 does not report any descriptive statistics, please revise its caption.

Response 7: Thank you for your valuable suggestion and we have revised this statement in the revised manuscript. The revised content is shown as follow:

Page 5 Lines 243-244: Table 1 Information About Variables Properties: Number of Items, Response Options, and Coding Procedure

Point 8: I would add what you think the covid-19 pandemic may have changed. Do you think loneliness is greatly increased and social participation changed? The present results continue to be informative?

Response 8: Thank you for your prospective idea. Due to we couldn’t obtain the relevant data at present, we cannot provide an accurate conclusion about the effect of the COVID-19 pandemic. Nevertheless, we have added a brief description of the phenomenon as a limit in the Discussion section and you can find the detailed content below:

Page 14 Lines 484-491: Finally, due to the limitation of data available, this paper didn’t take the effect of COVID-19 pandemic into consideration. Some relevant researches have shown that loneliness has become a global health concern caused by reduced social contacts and activities due to enforced restrictions, such as lockdowns, self-quarantine, stay-at-home, and keeping a social distance, particularly for older adults [66]. Given the ongoing cat-astrophic effect of the COVID-19 pandemic on global health, the future work would further investigate and compare the relationship pattern among social participation, social support and loneliness before and after the COVID-19 pandemic.

Point 9: I would also discuss the possibility of deep investigation of the relationship between social participation, social support, and loneliness considering other individual characteristics, e.g. personality traits.

Response 9: Thank you for your suggestive comment. Accordingly, we have provided a description of the potential factors that linked with the relationship in the Introduction section due to the lack of our data releted to personality traits. The revised information can be found below:

Page 3 lines 92-98: The best effect of participation type and frequency on loneliness may be related to in-dividual characteristics such as age, socio-economic status, and personality traits. Former studies relevant to personality traits in older adults reported that neuroticism and extraversion were the most influential personality factors [27]. Specifically, for those who were active and extraverted, high frequency of social participation are conductive to lower loneliness, while for those who suffered from social disorders (i.e., anxiety about talking to others in public) or enjoy being alone, an infrequent participation may be better.

Page 3 lines 131-135: the association among social participation, social support and loneliness may vary with individual differences, including age, gender, marital status, living arrangement, health status, and economic conditions [1,3,8,9]. For example, some previous studies showed that compared with those older adults who lived with children, the empty-nesters older adults were more likely to benefit from various social activities [1].

Point 10: I would also mention more explicitly some limitations of the present study. For instance, I suggest considering that you did not have any measure of wellbeing. This lack should be discussed. What if increased loneliness did not reflect decreased wellbeing? Maybe this is not the picture, based on literature, but a discussion of this could be added.

Response 10: Thank you for your constructive suggestion. We are sorry that we cannot provide an accuracy discussion of wellbeing due to the lack of data, as well as beyond our research focus at present and we think it an interesting issue worth in-depth study. So, we would regard it as a limitation improved in the future.

Page 13 Lines 476-484: Third, in addition to social support, there may be other related factors (i.e., subjective well-being [11,45], depression [1,38], personality traits [27]) moderating or mediating the relationship between social participation and loneliness, leading to the results of this paper may only reflect part of the overall impact of social participation. Therefore, a database with more comprehensive indicators and finer dimensions is needed in the future to supplement the specific approaches for studying the relationship between social participation and loneliness, so as to provide a rigorous and rich theoretical framework and model overview.

Reviewer 4 Report

Thank you for the opportunity to review the manuscript focusing on the association between social participation and loneliness of Chinese elderly over time – the mediation effect of social support.

It is evident that the authors are not first language English speakers. They have engaged in complex analyses but due to language barriers, it is not adequately conveyed.

The Introduction needs to be expanded to account for the changing social context that may have contributed to heightened loneliness in 2018 compared to earlier years.

Please verify that it is not the same respondents who completed the same survey across the three waves (2013, 2015 and 2018). It appears to be different cohorts of respondents who completed the survey over three different time periods.  

The authors need to state clearly that they are focusing on 3 waves (2013, 2015 and 2018) and not two samples (i.e. new born and earlier born).

The authors use a single item measure for loneliness but a complex theoretical construct such as loneliness cannot be measured with a single item. There are standardized scales such as the UCLA Loneliness scale that could have been used for the purpose of the study.

Similarly, the authors use a very basic measure for social support that is not well validated. There are a range of established instruments for measuring this construct.

The internal consistency reliability of the instruments has not been provided.

I query the necessity of providing the formula for the statistical analyses. A descriptive overview of the analytical tool would have better served the purpose.

Under Results: What is the significance of mentioning the average loneliness score for the elderly in 2015 (line 210)? There is no substantiation provided for the assumptions made in the first two paragraphs of this section.

Also, the authors cannot state that “with the increase of age, the level of loneliness increased” (line 211) because (i) it is not the same sample (ii) the age difference between the cohorts are negligible (67.91; 68.03; 68,57).

My suggestion is that their assumptions would carry more weight if it was presented after the statistical test.

The authors need to focus their analyses and description on different cohorts not different age groups.

With reference to Fig 1 and Fig 2, the authors need to decide which graph actually speaks to the research question. It is Fig 1.

Line 253 does not make sense “We fit our HLM with whether or not, frequency…”

Line 254: it is actually Model 3 and not Model 1. Correction needed.

Lines 254-262: Model 3 examines loneliness with social participation but the authors report it almost as a univariate relationship (i.e. mean loneliness in 2015 is higher than baseline with no reference to social participation).

Line 281: “role in the relationship between them”. It is not appropriate academic language to refer to variables as “them”. Rather state “in the relationship between social participation and loneliness”.

Line 285-286: It is sufficient to say that the effect was significant if the CI does not contain zero.

Author Response

Response to Reviewer 4 Comments (you can also find it in the attachment)

Thank you for your valuable and insightful comments. We have provided the responses to all your comments in two ways: (i) as for some individual comments, we response them one by one; (ii) as for other comments that related to one issue, we combined them and provided several unified answers.

Here my responses:

It is evident that the authors are not first language English speakers. They have engaged in complex analyses but due to language barriers, it is not adequately conveyed.

Point 1: The Introduction needs to be expanded to account for the changing social context that may have contributed to heightened loneliness in 2018 compared to earlier years.

Response1: Thank you for your constructive suggestions. We have supplemented the changing social context in the Introduction and Discussion sections to account for make a deeper analysis about the exacerbate loneliness among Chinese older adults. The part of revised content could be shown as follows:

[Introduction]

Page 3 Line 111-122: Within the context of traditional family structure and collective culture in China, older adults tend to put the family member first, especially children, then friends and peers [23]. However, the last 30 years have seen drastic declines in fertility, diluted filial piety, and uneven rates of economic mobility, all of which have contributed to rapid increases in empty-nest households and in the proportions of left-behind older adults whose adult children leave home for employment, more family fragmentation and smaller family size prevent older adults from receiving family support, thus, they fail to nurture their desired family relationship. In order to compensate for the loss of close family attachment and to alleviate the potential loneliness, they attempt to seek alternative sources of support. Hence social participation become one of the important channels for them to substitute family network and a convoy of late-life social new network [30].

[Discussion]

Page 12 lines 404-419: The results of HLM unconditional average model and unconditional growth model indicated that from 2013 to 2018, the loneliness of the Chinese older adults was at a medium level and showed an increasing trend, … However, the dual attributes of age effect and cohort effect of loneliness among the Chinese older adults may lie in unique social and cultural factors. It’s not difficult to find the included older adults experienced the period when Chinese Family Planning Policy was implemented most strictly (1980s and 1990s), leading to increasing quantity of nuclear families (i.e., only one-child per family) and decreasing family size recently. Meanwhile, in the larger social context of accelerated urbanization which increased from 53.73% in 2013 to 59.58% in 2018 [57], more and more young adults have migrated from home for employment and physical distance between generations also has become much longer. …

Page 13 lines 427-440: Secondly, after verifying the time-varying characteristics of loneliness, … Compared with the older adults who participated in social activities with low frequency, the older adults with medium and high frequency had lower loneliness. Moreover, we also found that the proportion of older adults who engaged in social participation declined over time, as well as the frequency, which may be linked with the rapid digitalization construction since 2014. The data showed that the number of Chinese older internet users had increased from 11.7 million in 2013 to 54.7 million in 2018 [59]. …

Point 2: Please verify that it is not the same respondents who completed the same survey across the three waves (2013, 2015 and 2018). It appears to be different cohorts of respondents who completed the survey over three different time periods.  

Point 3: The authors need to state clearly that they are focusing on 3 waves (2013, 2015 and 2018) and not two samples (i.e. new born and earlier born).

Responses 2-3: Thank you for your above two comment and we are sorry that the ambiguous statements confused you. We have revised it and presented it as follow:

Page 4 lines 192-197: our final sample size was 25192 which including 7208, 8381, and 9063 respondents in 2013, 2015, and 2018, respectively. There were 12232 independent respondents, and the proportions of respondents who participated in one wave, two waves, and three waves were 30%, 33%, and 37%, respectively.

Point 4: The authors use a single item measure for loneliness but a complex theoretical construct such as loneliness cannot be measured with a single item. There are standardized scales such as the UCLA Loneliness scale that could have been used for the purpose of the study.

Point 5: Similarly, the authors use a very basic measure for social support that is not well validated. There are a range of established instruments for measuring this construct.

Point 6: The internal consistency reliability of the instruments has not been provided.

Responses 4-6: Thank you for your valuable comments and we are sorry for the error in the original manuscript. Considering that the above 3 questions were all about the variables and instruments, we combined them and provided a unified answer. We have explained the reasons for why we chose these instruments and you can find the detailed content below. Nevertheless, we also pointed out the potential limitation of our measurements that should be improved in the future work. The internal consistency reliability was waived for this study due to these single-item instruments.

Page 4 lines 202-214:

3.2 Measurements

3.2.1 Loneliness

Loneliness was assessed using a single-item from the Centre for Epidemiological Studies scale CES-D, which assesses the frequency of feeling lonely in the previous week [28]. Respondents were asked to rate the item on a 4-point Likert scale, ranging from 1 (“rarely or none of the time”) to 4 (“most or all of the time”). The higher the score was, the higher the loneliness was. We chose this single-item measure of loneliness for two reasons: first, there is no longitudinal and nationwide survey of older Chinese which used the standardized scale (i.e., ULCA). Second, different from the special researches on the structure, differential experience of loneliness in the field of psychology, this study focused on the global perception of loneliness, and this single-item measure has been demonstrated correlated highly with multi-item loneliness scales and widely used in previous studies [42-45].

Page 5 lines 226-231:

3.2.3 Social support

Social support was measured by the items “Suppose … time?” and ”What is … persons?”, which was well accepted by the Chinese older adults samples [34].

Point 7: I query the necessity of providing the formula for the statistical analyses. A descriptive overview of the analytical tool would have better served the purpose.

Response 7: Thank you for your comment and suggestion. We think it’s much clearer to provide the formula for the statistical analyses combined with a descriptive overview, so we choose to retain this section.

[Under Results]

Point 8: What is the significance of mentioning the average loneliness score for the elderly in 2015 (line 210)? There is no substantiation provided for the assumptions made in the first two paragraphs of this section.

Point 9: Also, the authors cannot state that “with the increase of age, the level of loneliness increased” (line 211) because (i) it is not the same sample (ii) the age difference between the cohorts are negligible (67.91; 68.03; 68,57).

Responses 8-9: Thank you for your comments and we are sorry for the vague statement. Now we provide our answers to your question: The first thing to explain is that 1.58 didn’t refer specifically to the average loneliness score in 2015, rather than the average score during 3 waves. Then, the results of HLM for the longitudinal data could address the growth trends of psychological indicators or behaviors from the perspective of statistics. In parallel, we conducted ANOVA to examine loneliness change among those who participated in all three survey waves (N=4480) and the result (2013: 1.45, 2015: 1.57, 2018: 1.68; F=58.878, p<.001) demonstrates our conclusion in the original manuscript.[note that the result of ANOVA wasn’t present in the manuscript, if you need it please inform us] As for the comment (ii), the data we used in this paper was from a longitudinal tracking survey where samples in each wave must be representative of the whole Chinese aged 45 or older, so the age difference was negligible which wasn’t our research focus. Therefore, we think it’s enough to support our assumptions.

Point 12: With reference to Fig 1 and Fig 2, the authors need to decide which graph actually speaks to the research question. It is Fig 1.

Response 12: Thank you for your insightful comment and reminder. We re-analyzed this section and readjusted the figures in the revised manuscript. Specifically, we retained the Fig 3 and revised its caption. Also, we added the Fig.2 to presented the cohort trends.

Page 8 lines 334-337:

Fig. 2 Cohort trends in loneliness by age group

Fig. 3 Age trends in loneliness by survey time point

Point 13: Line 253 does not make sense “We fit our HLM with whether or not, frequency…”

Point 14: Line 254: it is actually Model 3 and not Model 1. Correction needed.

Responses 13-14: Thank you for your valuable suggestions and we have readjusted them in the revised manuscript. Below is the revised content:

Page 9 lines 357-358: Table 4 displayed the HLM results about the effects of all three social participation indicators on loneliness. Model 3 examined the second research question.

Point 10: My suggestion is that their assumptions would carry more weight if it was presented after the statistical test.

Point 11: The authors need to focus their analyses and description on different cohorts not different age groups.

Point 15: Lines 254-262: Model 3 examines loneliness with social participation but the authors report it almost as a univariate relationship (i.e. mean loneliness in 2015 is higher than baseline with no reference to social participation).

Responses 10,11,15: Thank you for the above 3 comments which were all focused on data analyses and results explanation, we will provide an integrated answer. Actually, we have reported the association between social participation and loneliness in the original manuscript and would present it for you below in order that you can find and check it. Additionally, we analyzed Model 3 from 4 aspects that including the associations of loneliness with social participation, survey point, age group and other socio-demographic factors. The part of detailed information will be shown as follows:

Page 9 lines 358-361: In each survey point, after controlling for potential confounding variables, social participation was significantly negatively correlated with loneliness, and loneliness of older adults with social participation was 0.06 points lower than that of non-participation (S.E. = 0.013, p <0.001).

Point 16: Line 281: “role in the relationship between them”. It is not appropriate academic language to refer to variables as “them”. Rather state “in the relationship between social participation and loneliness”.

Response 16: Thank you for your suggestion and we have revised it:

Page 11 lines 382-385: Combining Model 5 with Model 7, … in the relationship between social participation and loneliness.

Point 17: Line 285-286: It is sufficient to say that the effect was significant if the CI does not contain zero.

Response 17: Thank you for your suggestion and we have revised it:

Page 12 lines 388-390: Table 6 showed the significance test results and we could find that the mediation effect was statistically significant because the 95% confidence interval didn’t contain 0.

Round 2

Reviewer 1 Report

The comments have been addressed by the authors. Thank you.

Reviewer 2 Report

Dear Editor,

The vast majority of my questions/suggestions were accepted by the authors, who provided substantial revisions to the manuscript. I believe this version is acceptable for publication.

Reviewer 3 Report

The authors have answered to all my previous issues.

Reviewer 4 Report

I have now reviewed the manuscript and can confirm that the authors have addressed my feedback satisfactorily.